# The Human Kernel

**Andrew Gordon Wilson**
CMU

**Christoph Dann**
CMU

**Christopher G. Lucas**
University of Edinburgh

**Eric P. Xing**
CMU

## Abstract

Bayesian nonparametric models, such as Gaussian processes, provide a compelling framework for automatic statistical modelling: these models have a high degree of flexibility, and automatically calibrated complexity. However, automating human expertise remains elusive; for example, Gaussian processes with standard kernels struggle on function extrapolation problems that are trivial for human learners. In this paper, we create function extrapolation problems and acquire human responses, and then design a kernel learning framework to reverse engineer the inductive biases of human learners across a set of behavioral experiments. We use the learned kernels to gain psychological insights and to extrapolate in human-like ways that go beyond traditional stationary and polynomial kernels. Finally, we investigate Occam's razor in human and Gaussian process based function learning.

## 1 Introduction

Truly intelligent systems can learn and make decisions without human intervention. Therefore it is not surprising that early machine learning efforts, such as the perceptron, have been neurally inspired [1]. In recent years, probabilistic modelling has become a cornerstone of machine learning approaches [2, 3, 4], with applications in neural processing [5, 6, 3, 7] and human learning [8, 9].

From a probabilistic perspective, the ability for a model to automatically discover patterns and perform extrapolation is determined by its support (which solutions are a priori possible), and inductive biases (which solutions are a priori likely). Ideally, we want a model to be able to represent many possible solutions to a given problem, with inductive biases which can extract intricate structure from limited data. For example, if we are performing character recognition, we would want our support to contain a large collection of potential characters, accounting even for rare writing styles, and our inductive biases to reasonably reflect the probability of encountering each character [10].

The support and inductive biases of a wide range of probabilistic models, and thus the ability for these models to learn and generalise, is implicitly controlled by a *covariance kernel*, which determines the similarities between pairs of datapoints. For example, Bayesian basis function regression (including, e.g., all polynomial models), splines, and infinite neural networks, can all exactly be represented as a Gaussian process with a particular kernel function [11, 10, 12]. Moreover, the Fisher kernel provides a mechanism to reformulate probabilistic generative models as kernel methods [13].

In this paper, we wish to reverse engineer human-like support and inductive biases for function learning, using a Gaussian process (GP) based kernel learning formalism. In particular:

- We create new human function learning datasets, including novel function extrapolation problems and multiple-choice questions that explore human intuitions about simplicity and explanatory power, available at `http://functionlearning.com/`.
- We develop a statistical framework for kernel learning from the predictions of a model, conditioned on the (training) information that model is given. The ability to sample *multiple* sets of posterior predictions from a model, at any input locations of our choice, given any dataset of our choice, provides unprecedented statistical strength for kernel learning. By contrast, standard kernel learning involves fitting a kernel to a fixed dataset that can only be viewed as a *single* realisation from a stochastic process. Our framework leverages *spectral mixture kernels* [14] and non-parametric estimates.

- We exploit this framework to directly learn kernels from human responses, which contrasts with all prior work on human function learning, where one compares a fixed model to human responses. Further, we consider individual rather than averaged human extrapolations.
- We interpret the learned kernels to gain scientific insights into human inductive biases, including the ability to adapt to new information for function learning. We also use the learned "human kernels" to inspire new types of covariance functions which can enable extrapolation on problems which are difficult for conventional GP models.
- We study Occam's razor in human function learning, and compare to GP marginal likelihood based model selection, which we show is biased towards under-fitting.
- We provide an expressive quantitative means to compare existing machine learning algorithms with human learning, and a mechanism to directly infer human prior representations.

Our work is intended as a preliminary step towards building probabilistic kernel machines that encapsulate human-like support and inductive biases. Since state of the art machine learning methods perform conspicuously poorly on a number of extrapolation problems which would be easy for humans [12], such efforts have the potential to help automate machine learning and improve performance on a wide range of tasks – including settings which are difficult for humans to process (e.g., big data and high dimensional problems). Finally, the presented framework can be considered in a more general context, where one wishes to efficiently reverse engineer interpretable properties of any model (e.g., a deep neural network) from its predictions.

We further describe related work in section 2. In section 3 we introduce a framework for learning kernels from human responses, and employ this framework in section 4. In the supplement, we provide background on Gaussian processes [11], which we recommend as a review.

## 2  Related Work

Historically, efforts to understand human function learning have focused on rule-based relationships (e.g., polynomial or power-law functions) [15, 16], or interpolation based on similarity learning [17, 18]. Griffiths et al. [19] were the first to note that a Gaussian process framework can be used to unify these two perspectives. They introduced a GP model with a mixture of RBF and polynomial kernels to reflect the human ability to learn arbitrary smooth functions while still identifying simple parametric functions. They applied this model to a standard set of evaluation tasks, comparing predictions on simple functions to averaged human judgments, and interpolation performance to human error rates. Lucas et al. [20, 21] extended this model to accommodate a wider range of phenomena, and to shed light on human predictions given sparse data.

Our work complements these pioneering Gaussian process models and prior work on human function learning, but has many features that distinguish it from previous contributions: (1) rather than iteratively building models and comparing them to human predictions, based on fixed assumptions about the regularities humans can recognize, we are directly *learning* the properties of the human model through advanced kernel learning techniques; (2) essentially all models of function learning, including past GP models, are evaluated on averaged human responses, setting aside individual differences and erasing critical statistical structure in the data[1]. By contrast, our approach uses individual responses; (3) many recent model evaluations rely on relatively small and heterogeneous sets of experimental data. The evaluation corpora using recent reviews [22, 19] are limited to a small set of parametric forms, and more detailed analyses tend to involve only linear, quadratic and logistic functions. Other projects have collected richer data [23, 24], but we are only aware of coarse-grained, qualitative analyses using these data. Moreover, experiments that depart from simple parametric functions tend to use very noisy data. Thus it is unsurprising that participants tend to revert to the prior mode that arises in almost all function learning experiments: linear functions, especially with slope-1 and intercept-0 [23, 24] (but see [25]). In a departure from prior work, we create original function learning problems with no simple parametric description and no noise – where it is obvious that human learners cannot resort to simple rules – and acquire the human data ourselves. We hope these novel datasets will inspire more detailed findings on function learning; (4) we learn kernels from human responses, which (i) provide insights into the biases driving human function learning and the human ability to progressively adapt to new information, and (ii) enable human-like extrapolations on problems that are difficult for conventional GP models; and (5) we investigate Occam's razor in human function learning and nonparametric model selection.

# 3 The Human Kernel

The rule-based and associative theories for human function learning can be unified as part of a Gaussian process framework. Indeed, Gaussian processes contain a large array of probabilistic models, and have the non-parametric flexibility to produce infinitely many consistent (zero training error) fits to any dataset. Moreover, the support and inductive biases of a GP are encaspulated by a covariance kernel. Our goal is to learn GP covariance kernels from predictions made by humans on function learning experiments, to gain a better understanding of human learning, and to inspire new machine learning models, with improved extrapolation performance, and minimal human intervention.

## 3.1 Problem Setup

A (human) learner is given access to data $\mathbf{y}$ at training inputs $X$, and makes predictions $\mathbf{y}_*$ at testing inputs $X_*$. We assume the predictions $\mathbf{y}_*$ are samples from the learner's posterior distribution over possible functions, following results showing that human inferences and judgments resemble posterior samples across a wide range of perceptual and decision-making tasks [26, 27, 28]. We assume we can obtain multiple draws of $\mathbf{y}_*$ for a given $X$ and $\mathbf{y}$.

## 3.2 Kernel Learning

In standard GP applications, one has access to a single realisation of data $\mathbf{y}$, and performs kernel learning by optimizing the marginal likelihood of the data with respect to covariance function hyper-parameters $\boldsymbol{\theta}$ (supplement). However, with only a single realisation of data we are highly constrained in our ability to learn an expressive kernel function – requiring us to make strong assumptions, such as RBF covariances, to extract useful information from the data. One can see this by simulating $N$ datapoints from a GP with a known kernel, and then visualising the empirical estimate $\mathbf{y}\mathbf{y}^\top$ of the known covariance matrix $K$. The empirical estimate, in most cases, will look nothing like $K$. However, perhaps surprisingly, if we have even a small number of *multiple draws* from a GP, we can recover a wide array of covariance matrices $K$ using the empirical estimator $YY^\top/M - \bar{\mathbf{y}}\bar{\mathbf{y}}^\top$, where $Y$ is an $N \times M$ data matrix, for $M$ draws, and $\bar{\mathbf{y}}$ is a vector of empirical means.

The typical goal in choosing kernels is to use training data to find one that minimizes some loss function, e.g., generalisation error, but here we want to reverse engineer the kernel of a model – here, whatever model human learners are tacitly using – that has been applied to training data, based on both training data and predictions of the model. If we have a single sample extrapolation, $\mathbf{y}_*$, at test inputs $X_*$, based on training points $\mathbf{y}$, and Gaussian noise, the probability $p(\mathbf{y}_*|\mathbf{y}, k_{\boldsymbol{\theta}})$ is given by the posterior predictive distribution of a Gaussian process, with $\mathbf{f}_* \equiv \mathbf{y}_*$. One can use this probability as a utility function for kernel learning, much like the marginal likelihood. See the supplement for details of these distributions.

Our problem setup affords unprecedented opportunities for flexible kernel learning. If we have multiple sample extrapolations from a given set of training data, $\mathbf{y}_*^{(1)}, \mathbf{y}_*^{(2)}, \ldots, \mathbf{y}_*^{(W)}$, then the predictive conditional marginal likelihood becomes $\prod_{j=1}^{W} p(\mathbf{y}_*^{(j)}|\mathbf{y}, k_{\boldsymbol{\theta}})$. One could apply this new objective, for instance, if we were to view different human extrapolations as multiple draws from a common generative model. Clearly this assumption is not entirely correct, since different people will have different biases, but it naturally suits our purposes: we are not as interested in the differences between people, as the *shared* inductive biases, and assuming multiple draws from a common generative model provides extraordinary statistical strength for learning these shared biases. Ultimately, we will study both the differences and similarities between the responses.

One option for kernel learning is to specify a flexible parametric form for $k$ and then learn $\boldsymbol{\theta}$ by optimizing our chosen objective functions. For this approach, we choose the recent spectral mixture kernels of Wilson and Adams [14], which can model a wide range of stationary covariances, and are intended to help automate kernel selection. However, we note that our objective function can readily be applied to other parametric forms. We also consider empirical non-parametric kernel estimation, since non-parametric kernel estimators can have the flexibility to converge to *any* positive definite kernel, and thus become appealing when we have the signal strength provided by multiple draws from a stochastic process.

# 4 Human Experiments

We wish to discover kernels that capture human inductive biases for learning functions and extrapolating from complex or ambiguous training data. We start by testing the consistency of our kernel learning procedure in section 4.1. In section 4.2, we study progressive function learning. Indeed,

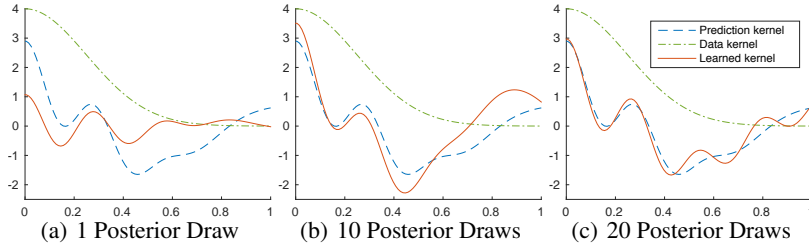

| Kernel |
|---|
| – – Prediction kernel |
| ·—· Data kernel |
| —— Learned kernel |

(a) 1 Posterior Draw    (b) 10 Posterior Draws    (c) 20 Posterior Draws

Figure 1: Reconstructing a kernel used for predictions: Training data were generated with an RBF kernel (green, ·—), and multiple independent posterior predictions were drawn from a GP with a spectral-mixture prediction kernel (blue, - -). As the number of posterior draws increases, the learned spectral-mixture kernel (red, —) converges to the prediction kernel.

humans participants will have a different representation (e.g., learned kernel) for different observed data, and examining how these representations progressively adapt with new information can shed light on our prior biases. In section 4.3, we learn human kernels to extrapolate on tasks which are difficult for Gaussian processes with standard kernels. In section 4.4, we study model selection in human function learning. All human participants were recruited using Amazon's mechanical turk and saw experimental materials provided at `http://functionlearning.com`. When we are considering stationary ground truth kernels, we use a spectral mixture for kernel learning; otherwise, we use a non-parametric empirical estimate.

### 4.1   Reconstructing Ground Truth Kernels

We use simulations with a known ground truth to test the consistency of our kernel learning procedure, and the effects of multiple posterior draws, in converging to a kernel which has been used to make predictions.

We sample 20 datapoints $\mathbf{y}$ from a GP with RBF kernel (the supplement describes GPs), $k_{\text{RBF}}(\mathbf{x}, \mathbf{x}') = \exp(-0.5||\mathbf{x} - \mathbf{x}'||/\ell^2)$, at random input locations. Conditioned on these data, we then sample multiple posterior draws, $\mathbf{y}_*^{(1)}, \ldots, \mathbf{y}_*^{(W)}$, each containing 20 datapoints, from a GP with a spectral mixture kernel [14] with two components (the prediction kernel). The prediction kernel has deliberately not been trained to fit the data kernel. To reconstruct the prediction kernel, we learn the parameters $\boldsymbol{\theta}$ of a randomly initialized spectral mixture kernel with five components, by optimizing the predictive conditional marginal likelihood $\prod_{j=1}^{W} p(\mathbf{y}_*^{(j)}|\mathbf{y}, k_{\boldsymbol{\theta}})$ wrt $\boldsymbol{\theta}$.

Figure 1 compares the learned kernels for different numbers of posterior draws $W$ against the data kernel (RBF) and the prediction kernel (spectral mixture). For a single posterior draw, the learned kernel captures the high-frequency component of the prediction kernel but fails at reconstructing the low-frequency component. Only with multiple draws does the learned kernel capture the longer-range dependencies. The fact that the learned kernel converges to the *prediction kernel*, which is different from the *data kernel*, shows the consistency of our procedure, which could be used to infer aspects of human inductive biases.

### 4.2   Progressive Function Learning

We asked humans to extrapolate beyond training data in two sets of 5 functions, each drawn from GPs with known kernels. The learners extrapolated on these problems in sequence, and thus had an opportunity to progressively learn about the underlying kernel in each set. To further test progressive function learning, we repeated the first function at the end of the experiment, for six functions in each set. We asked for *extrapolation* judgments because they provide more information about inductive biases than interpolation, and pose difficulties for conventional GP kernels [14, 12, 29].

The observed functions are shown in black in Figure 2, the human responses in blue, and the true extrapolation in dashed black. In the first two rows, the black functions are drawn from a GP with a rational quadratic (RQ) kernel [11] (for heavy tailed correlations); there are 20 participants.

We show the learned human kernel, the data generating kernel, the human kernel learned from a spectral mixture, and an RBF kernel trained only on the data, in Figures 2(g) and 2(h), respectively corresponding to Figures 2(a) and 2(f). Initially, both the human learners and RQ kernel show heavy tailed behaviour, and a bias for decreasing correlations with distance in the input space, but the human learners have a high degree of variance. By the time they have seen Figure 2(h), they are

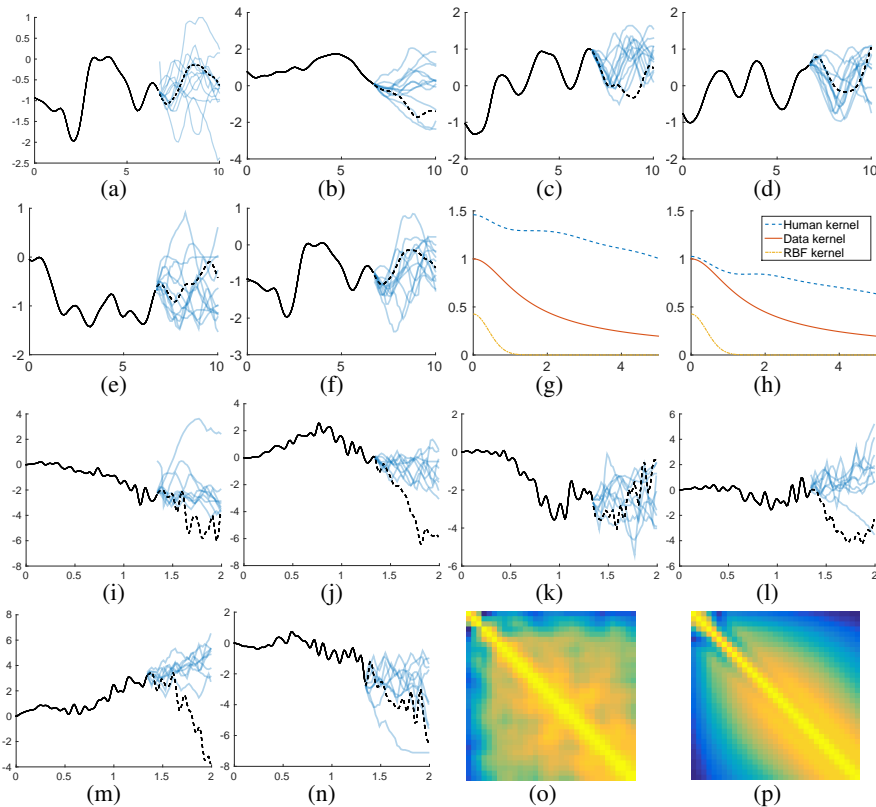

Figure 2: Progressive Function Learning. Humans are shown functions in sequence and asked to make extrapolations. Observed data are in black, human predictions in blue, and true extrapolations in dashed black. (a)-(f): observed data are drawn from a rational quadratic kernel, with identical data in (a) and (f). (g): Learned human and RBF kernels on (a) alone, and (h): on (f), after seeing the data in (a)-(e). The true data generating rational quadratic kernel is shown in red. (i)-(n): observed data are drawn from a product of spectral mixture and linear kernels with identical data in (i) and (n). (o): the empirical estimate of the human posterior covariance matrix from all responses in (i)-(n). (p): the true posterior covariance matrix for (i)-(n).

more confident in their predictions, and more accurately able to estimate the true signal variance of the function. Visually, the extrapolations look more confident and reasonable. Indeed, the human learners will adapt their representations (e.g., learned kernels) to different datasets. However – although the human learners will adapt their representations (e.g., learned kernels) to observed data – we can see in Figure 2(f) that the human learners are still over-estimating the tails of the kernel, perhaps suggesting a strong prior bias for heavy-tailed correlations.

The learned RBF kernel, by contrast, cannot capture the heavy tailed nature of the training data (long range correlations), due to its Gaussian parametrization. Moreover, the learned RBF kernel underestimates the signal variance of the data, because it overestimates the noise variance (not shown), to explain away the heavy tailed properties of the data (its model misspecification).

In the second two rows, we consider a problem with highly complex structure, and only 10 participants. Here, the functions are drawn from a product of spectral mixture and linear kernels. As the participants see more functions, they appear to expect linear trends, and become more similar in their predictions. In Figures 2(o) and 2(p), we show the learned and true predictive correlation matrices using empirical estimators which indicate similar correlation structure.

## 4.3 Discovering Unconventional Kernels

The experiments reported in this section follow the same general procedure described in Section 4.2. In this case, 40 human participants were asked to extrapolate from two single training sets, in counterbalanced order: a sawtooth function (Figure 3(a)), and a step function (Figure 3(b)), with traing data showing as dashed black lines.

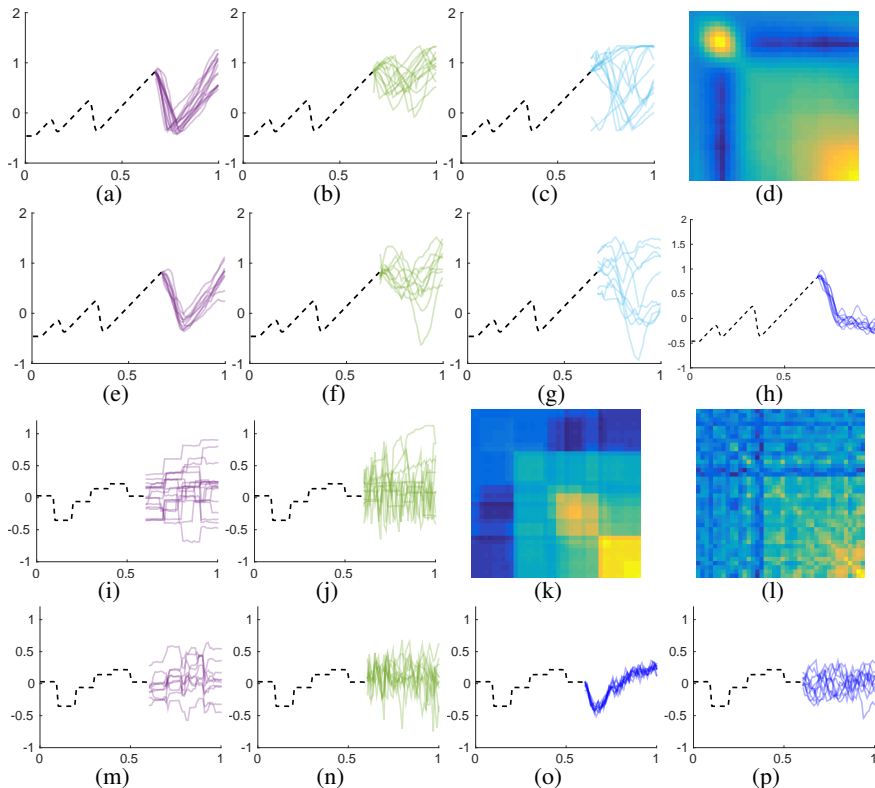

Figure 3: Learning Unconventional Kernels. (a)-(c): sawtooth function (dashed black), and three clusters of human extrapolations. (d) empirically estimated human covariance matrix for (a). (e)-(g): corresponding posterior draws for (a)-(c) from empirically estimated human covariance matrices. (h): posterior predictive draws from a GP with a spectral mixture kernel learned from the dashed black data. (i)-(j): step function (dashed black), and two clusters of human extrapolations. (k) and (l) are the empirically estimated human covariance matrices for (i) and (j), and (m) and (n) are posterior samples using these matrices. (o) and (p) are respectively spectral mixture and RBF kernel extrapolations from the data in black.

These types of functions are notoriously difficult for standard Gaussian process kernels [11], due to sharp discontinuities and non-stationary behaviour. In Figures 3(a), 3(b), 3(c), we used agglomerative clustering to process the human responses into three categories, shown in purple, green, and blue. The empirical covariance matrix of the first cluster (Figure 3(d)) shows the dependencies of the sawtooth form that characterize this cluster. In Figures 3(e), 3(f), 3(g), we sample from the learned human kernels, following the same colour scheme. The samples appear to replicate the human behaviour, and the purple samples provide reasonable extrapolations. By contrast, posterior samples from a GP with a spectral mixture kernel trained on the black data in this case quickly revert to a prior mean, as shown in Fig 3(h). The data are sufficiently sparse, non-differentiable, and non-stationary, that the spectral mixture kernel is less inclined to produce a long range extrapolation than human learners, who attempt to generalise from a very small amount of information.

For the step function, we clustered the human extrapolations based on response time and total variation of the predicted function. Responses that took between 50 and 200 seconds and did not vary by more than 3 units, shown in Figure 3(i), appeared reasonable. The other responses are shown in Figure 3(j). The empirical covariance matrices of both sets of predictions in Figures 3(k) and 3(l) show the characteristics of the responses. While the first matrix exhibits a block structure indicating step-functions, the second matrix shows fast changes between positive and negative dependencies characteristic for the high-frequency responses. Posterior sample extrapolations using the empirical human kernels are shown in Figures 3(m) and 3(n). In Figures 3(o) and 3(p) we show posterior samples from GPs with spectral mixture and RBF kernels, trained on the black data (e.g., given the same information as the human learners). The spectral mixture kernel is able to extract some structure (some horizontal and vertical movement), but is overconfident, and unconvincing compared to the human kernel extrapolations. The RBF kernel is unable to learn much structure in the data.

## 4.4 Human Occam's Razor

If you were asked to predict the next number in the sequence $9, 15, 21, \ldots$, you are likely more inclined to guess $27$ than $149.5$. However, we can produce either answer using different hypotheses that are entirely consistent with the data. *Occam's razor* describes our natural tendency to favour the simplest hypothesis that fits the data, and is of foundational importance in statistical model selection. For example, MacKay [30] argues that Occam's razor is automatically embodied by the marginal likelihood in performing Bayesian inference: indeed, in our number sequence example, marginal likelihood computations show that $27$ is millions of times more probable than $149.5$, even if the prior odds are equal.

Occam's razor is vitally important in nonparametric models such as Gaussian processes, which have the flexibility to represent infinitely many consistent solutions to any given problem, but avoid overfitting through Bayesian inference. For example, the marginal likelihood of a Gaussian process (supplement) separates into automatically calibrated model fit and model complexity terms, sometimes referred to as *automatic Occam's razor* [31].

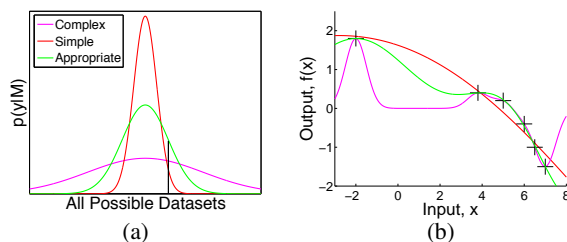

Figure 4: Bayesian Occam's Razor. a) The marginal likelihood (evidence) vs. all possible datasets. The dashed vertical line corresponds to an example dataset $\tilde{\mathbf{y}}$. b) Posterior mean functions of a GP with RBF kernel and too short, too large, and maximum marginal likelihood length-scales. Data are denoted by crosses.

The marginal likelihood $p(\mathbf{y}|\mathcal{M})$ is the probability that if we were to randomly sample parameters from $\mathcal{M}$ that we would create dataset $\mathbf{y}$ [e.g., 31]. Simple models can only generate a small number of datasets, but because the marginal likelihood must normalise, it will generate these datasets with high probability. Complex models can generate a wide range of datasets, but each with typically low probability. For a given dataset, the marginal likelihood will favour a model of more appropriate complexity. This argument is illustrated in Fig 4(a). Fig 4(b) illustrates this principle with GPs.

Here we examine Occam's razor in human learning, and compare the Gaussian process marginal likelihood ranking of functions, all consistent with the data, to human preferences. We generated a dataset sampled from a GP with an RBF kernel, and presented users with a subsample of 5 points, as well as seven possible GP function fits, internally labelled as follows: (1) the predictive mean of a GP after maximum marginal likelihood hyperparameter estimation; (2) the generating function; (3-7) the predictive means of GPs with larger to smaller length-scales (simpler to more complex fits). We repeated this procedure four times, to create four datasets in total, and acquired 50 human rankings on each, for 200 total rankings. Each participant was shown the same unlabelled functions but with different random orderings.

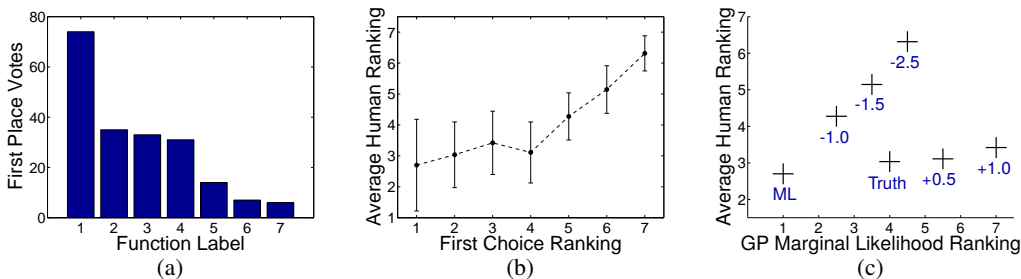

Figure 5: Human Occam's Razor. (a) Number of first place (highest ranking) votes for each function. (b) Average human ranking (with standard deviations) of functions compared to first place ranking defined by (a). (c) Average human ranking vs. average GP marginal likelihood ranking of functions. 'ML' = marginal likelihood optimum, 'Truth' = true extrapolation. Blue numbers are offsets to the log length-scale from the ML optimum. Positive offsets correspond to simpler solutions.

Figure 5(a) shows the number of times each function was voted as the best fit to the data, which follows the internal (latent) ordering defined above. The maximum marginal likelihood solution receives the most (37%) first place votes. Functions 2, 3, and 4 received similar numbers (between 15% and 18%) of first place votes. The solutions which have a smaller length-scale (greater complexity) than the marginal likelihood best fit – represented by functions 5, 6, and 7 – received a relatively small number of first place votes. These findings suggest that on average humans prefer overly simple explanations of the data. Moreover, participants generally agree with the GP marginal likelihood's first choice preference, even over the true generating function. However, these data also suggest that participants have a wide array of prior biases, leading to variability in first choice preferences. Furthermore, 86% (43/50) of participants responded that their first ranked choice was "likely to have generated the data" and looks "very similar" to imagined.

It's possible for highly probable solutions to be underrepresented in Figure 5(a): we might imagine, for example, that a particular solution is never ranked first, but always second. In Figure 5(b), we show the average rankings, with standard deviations (the standard errors are stdev/$\sqrt{200}$), compared to the first choice rankings, for each function. There is a general correspondence between rankings, suggesting that although human distributions over functions have different modes, these distributions have a similar allocation of probability mass. The standard deviations suggest that there is relatively more agreement that the complex small length-scale functions (labels 5, 6, 7) are improbable, than about specific preferences for functions 1, 2, 3, and 4.

Finally, in Figure 5(c), we compare the average human rankings with the average GP marginal likelihood rankings. There are clear trends: (1) humans agree with the GP marginal likelihood about the best fit, and that empirically decreasing the length-scale below the best fit value monotonically decreases a solution's probability; (2) humans penalize simple solutions *less* than the marginal likelihood, with function 4 receiving a last (7th) place ranking from the marginal likelihood.

Despite the observed human tendency to favour simplicity more than the GP marginal likelihood, Gaussian process marginal likelihood optimisation is surprisingly biased towards *under-fitting* in function space. If we generate data from a GP with a known length-scale, the mode of the marginal likelihood, on average, will over-estimate the true length-scale (Figures 1 and 2 in the supplement). If we are unconstrained in estimating the GP covariance matrix, we will converge to the maximum likelihood estimator, $\hat{K} = (\mathbf{y} - \bar{y})(\mathbf{y} - \bar{y})^{\top}$, which is degenerate and therefore biased. Parametrizing a covariance matrix by a length-scale (for example, by using an RBF kernel), restricts this matrix to a low-dimensional manifold on the full space of covariance matrices. A biased estimator will remain biased when constrained to a lower dimensional manifold, as long as the manifold allows movement in the direction of the bias. Increasing a length-scale moves a covariance matrix towards the degeneracy of the unconstrained maximum likelihood estimator. With more data, the low-dimensional manifold becomes more constrained, and less influenced by this under-fitting bias.

## 5 Discussion

We have shown that (1) human learners have systematic expectations about smooth functions that deviate from the inductive biases inherent in the RBF kernels that have been used in past models of function learning; (2) it is possible to extract kernels that reproduce qualitative features of human inductive biases, including the variable sawtooth and step patterns; (3) that human learners favour smoother or simpler functions, even in comparison to GP models that tend to over-penalize complexity; and (4) that is it possible to build models that extrapolate in human-like ways which go beyond traditional stationary and polynomial kernels.

We have focused on human extrapolation from noise-free nonparametric relationships. This approach complements past work emphasizing simple parametric functions and the role of noise [e.g., 24], but kernel learning might also be applied in these other settings. In particular, iterated learning (IL) experiments [23] provide a way to draw samples that reflect human learners' a priori expectations. Like most function learning experiments, past IL experiments have presented learners with sequential data. Our approach, following Little and Shiffrin [24], instead presents learners with plots of functions. This method is useful in reducing the effects of memory limitations and other sources of noise (e.g., in perception). It is possible that people show different inductive biases across these two presentation modes. Future work, using multiple presentation formats with the same underlying relationships, will help resolve these questions.

Finally, the ideas discussed in this paper could be applied more generally, to discover interpretable properties of unknown models from their predictions. Here one encounters fascinating questions at the intersection of active learning, experimental design, and information theory.

## Footnotes

[1]For example, averaging prior draws from a Gaussian process would remove the structure necessary for kernel learning, leaving us simply with an approximation of the prior mean function.

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
