[Supplementary Material]

# Supplementary Material: The Human Kernel

**Andrew Gordon Wilson**
CMU

**Christoph Dann**
CMU

**Christopher G. Lucas**
University of Edinburgh

**Eric P. Xing**
CMU

Supplementary materials are available at `http://www.cs.cmu.edu/~andrewgw/humansupp.pdf`, providing a brief review of Gaussian processes, and additional experiments regarding the under-fitting property of GP maximum marginal likelihood estimation of kernel length-scales. We also provide the instructions and some of the questions asked in the human experiments. To participate in the exact experiments, and additional resources, see `http://www.functionlearning.com`.