[Reviews · NeurIPS 2015]

Submitted by Assigned_Reviewer_1

I liked the general theme of the paper (i.e. learning kernel that encodes human biases). The authors have focused only on one-dimensional problems. I'm not sure how the current work would be extended to higher dimensions. The authors claim that the framework can be considered as a general framework to efficiently reverse engineer interpretable properties of models (lines 72-74). It'd be helpful to add a discussion.

I think this line of work is quite novel in the Gaussian process community. However, I am not an expert in psychology, so I'm not aware of work in that literature on modeling human biases. I'll leave it to cognitive science experts to assess originality.

Have you also thought about other kinds of biases e.g., symmetry? In section 2, the authors mention that they create original functions where the human learners cannot resort to simple rules. I'm not sure which extrapolations would count as "simple rules" and why exactly the authors ignored such cases; it'd be helpful to clarify this.

It was not very clear to me why (g) and (h) are different in Fig 2. Did the same users create different extrapolations in (a) and (f)? If so, did you ask them why they provided a different extrapolation?

Minor comments:

Line 298: (i)-(h) -> (i)-(j)? Line 304: not clear what "vary more than 3" means Fig 4: colors look very similar Line 87 supplementary material: Fig 2 -> Fig 1?
Summary: The authors describe a framework to learn kernels that reverse engineer human inductive biases. The paper is very well-written and I enjoyed reading it. The problem setup seems quite novel to me and the authors have collected an interesting dataset of human extrapolations which should inspire further work along these lines. Overall, I think this is a good paper that would be of interest to a wide NIPS audience.

Submitted by Assigned_Reviewer_2

Rebuttal: thank you for your clarifications. I still think that learning kernel (parameters) from multiple realizations of a GP is not very novel in general, but sufficiently novel in your specific context to get discussed at NIPS.

The authors use Gaussian processes to learn human function extrapolation behaviour from human sample data. After a comprehensive literature review, they introduce the main idea of the paper: learn the kernel parameters by maximizing the conditional probability of the extrapolation data given the training data. To allow for flexible kernel shapes, they use spectral mixture kernels. They show

by ground truth testing that these kernels can be learned. Then, they show that spectral mixture kernels can learn more reasonable approximations to generating functions in extrapolation tasks than RBF kernels when trained on human data. In section 4.3, it is demonstrated that functions with a non-stationary covariance functions are hard to learn. Finally, the authors study if Bayesian model selection in GPs is comparable to human model selection. They find that: a) GP model selection prefers models that are too simple when the number of datapoints is small and b) humans prefer even simpler functions. a) is unsurprising, b) is interesting. The paper is clearly written, the work is of sufficient quality for NIPS. It is definitely incremental and its significance in the context of a large body of previous work on human function extrapolation is modest.

Detailed comments:

line 54,146: the claim of "unprecedented statistical strength" is surprising. Learning kernels from multiple realizations of a GP is a standard procedure, see for example GP latent variable model learning (Lawrence 2004).

line 187: is there a square missing in the exponent of the RBF kernel?

Please specify the functional form of the spectral mixture kernels somewhere in section 3.2

figure 2, g+h: I found it surprising that the RBF kernel has a variance of only 0.5, when the data variance is about twice as large. Please explain.

line 298: (i)-(h) should be (i)-(j)?

figure 4a is superfluous. Instead, the legend of fig 4b needs to be improved. Have the colors been switched between figs 4a and 4b? I assume the crosses are the datapoints?

Summary: An incremenal contribution to a growing body of research trying to use human data for constraining Gaussian Process (GP) learning. The main novelty claimed by the authors is learning a shared kernel from multiple realizations of a GP, but that is a standard procedure for learning GP latent variable models (Lawrence 2004) with high-dimensional inputs.

Submitted by Assigned_Reviewer_3

The paper proposes a method for learning a kernel for Gaussian process modeling, that represents human decision making and data extrapolation as accurately as possible. Firstly a number of data sets are constructed, where human test persons have provided labels in the form of function extrapolations etc. The data points in connection to the human input concerning this data point, provide insight into human estimation and decision making. The approach in the paper is then to learn a kernel that in a GP framework imitates the human behavior. In contrast to standard kernel learning approaches, a Bayesian statistical framework is here developed that allows sampling of multiple posterior prodictions from the model, which allows for more efficient use of the data. The learned kernels are able to capture human behavior in reasoning e.g. about smooth functions in a much more accurate manner than RBF kernels.

Quality: This is a very well written, well organized, clear and convincing paper. A negative aspect is that many essential details about the experiments are left to the supplementary material, to such a degree that the experiments are not repeatable given only the information in the paper. However, this is understandable given the restrictive spatial constraints. The proposed kernel learning method is only evaluated on quite simplistic data, but the evaluation is valid and systematic, and the interpretation of the experiments is convincing.

Clarity: The reasoning in the paper is extremely clear. The contributions with respect to related work are made clear early on, and there is a good motivation for the presented work.

Originality: This is slightly outside my area of expertise, but to my knowledge this is highly original work.

Significance: As argued in the introduction, methods that can predict human extrapolation and prediction and decision making - problems that are very difficult to automate using current methods - will be highly beneficial for a wide range of problems. Therefore, the human kernel proposed in this paper is potentially a highly significant contribution.
Summary: This paper is highly original and very well written. The proposed kernel learning method is only evaluated on quite simplistic data, but the evaluation is valid and systematic.

Submitted by Assigned_Reviewer_4

The paper is fantastic in many manners and very well-written. I will focus on (hopefully) constructive criticism:There is too much packed into the paper. As such, there are some details that should be in the paper, but are not. For example, a reader should be able to replicate the modeling and human experiments based on reading the article alone, if she desired. I do not think I could do either based on only the article. Although the supplement helps, it does not help with the experiment presented in section 5. My main other criticism is that the psychological implications of the studies presented by the authors is sloppy at times. Does it really get at THE human kernel? Their own results seem to contradict this as different kernels were learned depending on the experiment (compare Figure 2o to Figure 3d). I also suspect the kernel will be highly domain-dependent. Also, the authors write "they are more confident in their predictions...". As far as I can tell, the researchers did not get individual confidence ratings and without that, it seems inappropriate to say that they are more confident. I would say that they is less variability between participants. Finally, I have one last concern with respect to the stimuli. As the authors note at the end, the nature of the stimuli typically used in the human function learning is very different than that used in the experiments. I'm not a fan of the standard human function learning stimuli, but you should mention that it is going to be different right away, as it is confusing to cognitive science readers that are familiar with the human function learning literature. Further, the author's stimulus presentation introduces a confound: they could be studying human inductive biases over 1-d curves in images rather than functions themselves. It is probably worth mentioning this limitation of this method.

More minor concerns: Paragraph 2 of 4.1: I don't understand what the prediction kernel is. Also, why use RBFs if later you focus on RQs? Also also, you never define RQ... I found Figure 2 difficult to parse. It might be worth considering other ways of organizing the subplots. If you are interested in human rule learning in number sequences, you might consider citing and/or taking a look at Austerweil & Griffiths (2011). Seeking Confirmation is Rational for Deterministic Hypotheses. Cognitive Science.

line 351-352: "Participant were" -> "Participants were" The sections for experiments are mislabeled in the supplementary materials (and also the last 2/3 experiments are missing).
Summary: An interesting study on human and machine function learning. The authors' experiments and results are provocative, and I expect they will be somewhat controversial and lead to interesting discussion.

Submitted by Assigned_Reviewer_5

This paper deals with an extremely interesting topic that is related to human cognition. I was really fascinated by the introductory part of the paper. Really regrettably, the problem setting described in Section 3 and 4 is not clear at all. The author should resubmit the paper with major revisions to make it more understandable.

Summary: This paper seems to propose a new kernel learning framework for Gaussian process regression so as to reproduce extrapolation capabilities characteristic of humans. The introductory part is impressive, but the formulation is unclear.

Author Feedback
Author rebuttal: We thank all reviewers for thoughtful comments, which are in general very positive about the quality, clarity, and originality of our work. Our work contains a variety of original contributions, enumerated below, and provides early steps towards building probabilistic kernel machines that encapsulate human-like inductive biases, which is of fundamental significance. We believe it will provoke discussion on foundational concepts. We respond to reviewers individually.

R1:

Thanks for your comments.

Typically kernel learning is performed from a single realisation of a Gaussian process (GP). As one aspect of our paper, we note that multiple realisations provide opportunities for ambitious kernel learning. Multiple realisations have been considered elsewhere, but in very different contexts, such as the GPLVM you mention. The GPLVM is used for unsupervised learning with high dimensional inputs, typically with simple kernels. By contrast, we are working in a supervised setting, where we wish to reverse engineer the properties of a model from its predictions given labelled training data, through a conditional marginal likelihood, and an expressive kernel learning framework, involving both spectral mixtures and non-parametric kernel estimators. The problem setup, model, modelling objectives, and scientific questions are all different, and it's not clear how the GPLVM could be applied to the problems we consider.

The original contributions of our paper include: (1) a distinctive problem setup, where we wish to reverse engineer the properties of a model from its predictions given labelled training data; (2) a new approach for this problem that makes it possible to learn expressive kernels; (3) exploiting this approach to directly learn kernels from human responses, which starkly contrasts with all prior work on human function learning, where one compares a fixed model to human responses; (4) new human function learning datasets, composed of non-parametric function extrapolation problems which are not standard for this experimental domain; (5) working with individual rather than averaged human responses; (6) interpreting the learned kernels to gain new psychological insights into human biases and our ability to progressively adapt to new information; (7) using the learned kernels to provide human-like extrapolations on fundamental problems which cannot be solved by conventional GP models; (8) investigating Occam's razor in human and GP based function learning.

We are glad that you found our results about humans preferring simpler models to GPs interesting. Although the point about marginal likelihood over-fitting causing GP function-space under-fitting may not be surprising to everyone, it is fundamental and largely unknown. In personal correspondence, other researchers have found this to be a highly intriguing result, with widespread implications for type-II maximum likelihood estimation, so we believe it could provoke interesting discussion in connection to our experiments.

Yes, there should be a square in the RBF kernel. In fig 2 g+h, the RBF kernel is underestimating the signal variance of the data because it is overestimating the noise variance, which we are not currently illustrating as part of the kernel. Yes, i-j. Yes, the colours were switched, and the crosses are data points. Thanks for these comments; we will accordingly make clarifications in a final version.

R2:

We are grateful for your thoughtful comments. We respond about experimental details to R3.

R3:

We really appreciate your support and constructive feedback. In the final version, we will include url links to the full interactive experiments, as well as links to datasets and code, so that every step is easy to replicate. As you have noted, the learned kernels from human responses will vary depending on the data, since the humans themselves are adapting their kernels to the data. We study these changes in the progressive function learning experiments. However, certain properties of the learned kernels remain relatively fixed, such as heavy tailed behaviour (e.g. fig 2, g+h), which indicate strong prior human biases. We will clarify this.

Yes, the experiments are quite different than what is standard in the human function learning literature, which we will mention earlier.

We will clarify the prediction kernel. We used the rational quadratic (RQ) kernel to generate heavy tailed functions. We will include your suggestions.

R4:

Thanks for the supportive feedback.

By simple rules we meant e.g. "this is a polynomial", rather than general non-parametric extrapolations.

2(g) and (h) are different because the extrapolations in 2(a) and (f) are different; when users saw 2(f) they had seen 2(a)-(e), which helped with learning about the function class. As a follow-up it would be interesting to ask users to elaborate on what caused their changed responses.